# *Brucella* Genomics: Macro and Micro Evolution

**DOI:** 10.3390/ijms21207749

**Published:** 2020-10-20

**Authors:** Marcela Suárez-Esquivel, Esteban Chaves-Olarte, Edgardo Moreno, Caterina Guzmán-Verri

**Affiliations:** 1Programa de Investigación en Enfermedades Tropicales, Escuela de Medicina Veterinaria, Universidad Nacional, Heredia 3000, Costa Rica; marcela.suarez.esquivel@una.cr (M.S.-E.); edgardo.moreno.robles@una.cr (E.M.); 2Centro de Investigación en Enfermedades Tropicales, Facultad de Microbiología, Universidad de Costa Rica, San José 1180, Costa Rica; esteban.chaves@ucr.ac.cr

**Keywords:** *Brucella*, brucellosis, genome reduction, pseudogene, IS*711*, SNPs

## Abstract

*Brucella* organisms are responsible for one of the most widespread bacterial zoonoses, named brucellosis. The disease affects several species of animals, including humans. One of the most intriguing aspects of the brucellae is that the various species show a ~97% similarity at the genome level. Still, the distinct *Brucella* species display different host preferences, zoonotic risk, and virulence. After 133 years of research, there are many aspects of the *Brucella* biology that remain poorly understood, such as host adaptation and virulence mechanisms. A strategy to understand these characteristics focuses on the relationship between the genomic diversity and host preference of the various *Brucella* species. Pseudogenization, genome reduction, single nucleotide polymorphism variation, number of tandem repeats, and mobile genetic elements are unveiled markers for host adaptation and virulence. Understanding the mechanisms of genome variability in the *Brucella* genus is relevant to comprehend the emergence of pathogens.

## 1. Introduction

The Proteobacteria phylum represents the most extensive bacteria domain known. This name is derived from the Greek *protean,* literary meaning “first,” and coincidentally to the god Proteus who could change his form. Within this phylum, there is the Alphaproteobacteria Class, encompassing bacteria with different shapes, lifestyles, metabolic capacities, and ecological variation (Figure 1) [1]. From the lifestyle perspective, this Class includes free-living, commensals, endosymbionts, opportunistic, and intracellular pathogenic bacteria [2,3,4] and therefore is an excellent model for studying the evolution of parasitic bacteria. The genus *Brucella* belongs to one of the Class families, the Brucellaceae family, which includes the genera *Pseudochrobactrum, Falsochrobactrum, Ochrobactrum*, and *Brucella* [5]. The average genome size of *Brucella* organisms is 3.3 Mb, divided into two circular chromosomes of different sizes: 2.1 Mb and 1.2 Mb [6]. There are *Brucella* organisms with a single megareplicon of 3.3 Mb, a consequence of the fusion of the two chromosomes [7]. The *Brucella* genome is conserved and stable due to traits such as its substantial DNA guanosine (G) + cytosine (C) content (~57% mol). Besides, there is no evidence of plasmids, horizontal DNA transference, active lysogenic phages [3], or recent external recombination events [3,8,9]. These properties, therefore, preclude the acquisition of exogenous genetic material by classical recombination or phage integration. Based on its genomic traits and strict molecular classification, *Brucella* is considered a monospecific genus. Nevertheless, the taxonomic ranking recognizes several species based on host range preference, phenotypic and genotypic characteristics, and biological behavior [10]. It is also clear that the classical *Brucella* organisms fit the definition of ecotypes from the biological, biochemical, genomic, and medical perspectives. It is intriguing how the various *Brucella* species/strains, displaying a very close genetic relationship (~97% similarity), still exhibit different host preferences [11], virulence [10,12], and zoonotic potential [13].

Information regarding differences in the genomic organization among *Brucella* species, phylogenetic proximity and the description of two separated chromosomes was obtained during the 1990s, using Pulse-Field Gel Electrophoresis (PFGE) [6,17,18]. Multi-locus sequence type (MLST) and multiple loci variable number of tandem repeats (VNTR) analysis (MLVA) are now used to infer genetic diversity in *Brucella* by sequencing multiple genetic loci. They allow geographic clustering of the isolates, tracking sources of infection and their global dispersion [1,2,3,4,5,6,7,8,9,10,11,12,13,14,15,16,17,18,19,20,21,22]. Although the MLST classification has some practical application, *sensu stricto* a genetic variability is a continuous event that cannot be restricted to types, but rather to phylogenetic relations. Therefore, MLST classification, as well as MLVA classification, have to be taken cautiously. Moreover, the branching and rooting topology of the dendrograms determined by MLVA and MLST do not necessarily coincide with the phylogenetic trees resolved by WGA.

The whole-genome sequence (WGS) analysis based on single nucleotide polymorphisms (SNPs) or core genome is currently the predominant technique to infer the evolutionary history and diversity of several pathogens in different hosts and locations, including *Brucella* species.

Following the increasing availability of *Brucella* WGS generated in the last decade and computational power to analyze these sequences [23,24,25,26,27], we have reviewed the variability of genetic traits concerning host preferences and the brucellae geographic origin.

## 2. Genome Reduction in Cell-Associated Alphaproteobacteria

Alphaproteobacteria organisms are found in different environments, including water and soil, and can form extra- and intracellular associations with unicellular and multicellular eukaryotes [28]. Environmental and plant-associated genera such as *Caulobacter* and *Bradyrhizobium*, possess large genomes ranging from 4.0 to 9.1 Mb, respectively [29,30]. With intermedium genome sizes between 3.3 and 1.6 Mb, there are intracellular-extracellular facultative organisms like *Brucella* and *Bartonella* species, which can infect animals and are zoonotic [3,7,31,32]. Taking apart mitochondria, *Rickettsia* and *Wolbachia* are the species with the smallest genome sizes of the Class, 1.1–1.3 Mb length, which are obligate intracellular pathogens or symbionts of arthropods and nematodes; some of them are zoonotic [33,34].

Gene loss and bacterial chromosomal streamlining are proposed mechanisms for generating smaller genomes [3,4,35]. Loss or reduction of non-essential genes or sequences might occur within the context of changes in the surrounding bacterial environment. In cases, gene reduction delivers the so-called “non-culturable” bacteria, which may be a predictive marker of pathogenicity in bacteria [36]. Examples of gene reduction are the intracellular parasitic bacteria of the order *Rickettsiales*, with genome size ranging from 2–1.1 Mb [28,37,38]. Some of the lost genes in this animal cell-associated bacteria are related to the synthesis of essential metabolites such as amino acids, carbohydrates, and vitamins supplied by the host cell [39]. *Wolbachia* species and mitochondria are extreme genome reduction cases, in which genome streamlining promoted a permanent stage of endosymbiosis, with a high replication rate within the host environment [4,35,40,41]. Likewise, there are reports in gene decay on critical metabolic pathways in *Brucella* [27,42], and the chromosomal streamlining seems to have been an essential step in the separation from its closest relative, *Ochrobactrum* [3,43], as further detailed.

## 3. *Brucella* Genome Macroevolution Includes Gene Loss

The close phylogenetic relationship of *Brucella* organisms with soil-arthropod-plant-associated bacteria suggests that the common ancestor of *Brucella* evolved from opportunistic organisms close to 500 million years ago that adapted to an intracellular life probably in a cold-blooded vertebrate [44], later on adapting to mammals [15,45,46]. Phylogenetically, the closest bacterial species to the genus *Brucella* are species of the genus *Ochrobactrum,* such as *Ochrobactrum anthropi* and *Ochrobactrum intermedium* (Figure 1). These are soil bacteria associated with plants that may behave as opportunistic bacteria causing nosocomial infections in immunocompromised human hosts [16,47]. Similar to *Brucella* species, *Ochrobactrum* organisms have two circular chromosomes. However, the genome size of *Ochrobactrum* is larger than the brucellae genome and contains several plasmids. The *Ochrobactrum anthropi* chromosome I contains ~2.9 million bp with a prototypical bacterial chromosome origin of replication and a chromosome II of ~1.9 million bp with a *repABC* origin commonly found in secondary chromosomes of a plasmid origin [48]. Both chromosomes have an average G + C content of ~56% and encode ~4425 protein-CDS, along with 31 pseudogenes and 73 structural RNAs (rRNA, tRNA, and small RNA). Due to their characteristics, it was proposed that the second smaller chromosome derived from a megaplasmid that acquired housekeeping genes [3]. Still, compared to other phylogenetically related Alphaproteobacteria associated with plants such as *Bradyrhizobium* or *Sinorhizobium* species, the genomes of *Ochrobactrum* organisms have undergone significant reduction, which may be a trait in this lineage [49].

As expected, the two *Ochrobactrum* sp. chromosomes are similar and display collinearity to the two brucellae chromosomes. Nevertheless, the brucellae genomes are approximately 30% smaller than those of *Ochrobactrum* organisms [3,28,48,50]. In at least one non-classical *Brucella* strain named *B. inopinata* BO1, a temperate-like broad host range phage, which under certain stringent conditions can be expressed, has been described [51]. Since this phage is similar to that found in *Ochrobactrum* ssp., it might indicate that ancestral brucellae harbor lysogenic phages. 

The close interaction during millions of years between *Brucella* organisms with host cells shaped the bacterial genome [52]. This condition likely imposed bottlenecks in the genes-flow of *Brucella*, favoring gene reduction and streamlining, including the elimination of accessory genetic elements [3,53]. The two main mechanisms of gene reduction and streamlining proposed in prokaryotes include deletion bias and purifying selection [41]. In the first mechanism, the rate of DNA loss is naturally higher than the rate of DNA acquisition [54]. In the second mechanism, DNA sequences are selected against, favoring the selective removal of deleterious alleles that are non-functional, non-essential, or redundant [55]. Alternatively, bacteria such as the obligate intracellular *Orientia tsutsugamushi*, display massive amplification of mobile elements during functional reductive genome evolution [38]. Close to 50% of the genome contains repetitive sequences derived from an integrative and conjugative element, ten types of transposable elements, and seven short repeats of unknown origins. The amplification and degradation of these sequences resulted in intensive genome shuffling and produced many repeated genes, most of them pseudogenes. 

Genome reduction can also stabilize selection through the purging of deleterious genetic polymorphisms that arise through random mutations. Deletion bias seems to be the mechanism responsible for restricting genome diversity and particularly relevant in those “clonal” prokaryotes in which recombination events are precluded, such as members of the genus *Brucella* [2]. Despite this, the genetic elements necessary for adapting to a parasitic way of life were retained and modified by internal genetic processes, such as mutation, inversion, translocation, and insertion/deletion of transposable elements, selected within the boundaries to the new restricted host cell environments. For instance, several orthologous structural scaffolds such as the VirB type IV secretion system, the lipopolysaccharide, cell envelope molecules, and various metabolic alternatives required for virulence and intracellular life were first acquired and then remodeled in the *Brucella* ancestor. For some of those, such as VirB, it has been also proposed that components of the system were horizontally transferred from different sources and times during *Brucella* evolution and virulence development [43,56]. Concomitantly to this event, a stealthy strategy for avoiding the host immune response was developed [57].

The acquisition of characteristics through gene rearrangement, duplication, mutation, and reorganization of mobile chromosomal elements such as insertion sequences (IS) within specific contexts, may have provided the genetic framework for establishing some of the *Brucella* intra-species diversity [58]. The ancestral acquisition of genes coding for proteins functioning in metabolism, degradation, biosynthesis, and metal transport clustered in genomic regions, called genomic islands (GIs), and shared anomalous regions (SARs); some of them are *Brucella* species-specific [59,60]. Others, such as genes coding for the O chain of lipopolysaccharide (LPS), functional flagella, phage truncated sequences, pili-like structures, and some broader metabolic alternatives (most of them non-essential for survival), vary between the so-called classical and non-classical *Brucella* groups [61] clustered in two phylogenetic groups (Figure 2) [62]. Still, other genes are orthologues to plant-associated Alphaprotebacteria but absent in *Ochrobactrum* species, suggesting that some of these islands and anomalous regions may have been present in the *Ochrobactrum/Brucella* ancestor but lost during the emergence of the *Ochrobactrum* lineage [43]. As stated, *Ochrobactrum* has also lost genes and reduced its genome in comparison to other Alphaproteobacteria close soil/plant relatives, such as *Agrobacterium* and *Sinorhizobium* organisms with larger genomes [49].

## 4. Expanding the Number of *Brucella* Species 

Based on WGS, the *Brucella* genus may be divided into two distinct phylogenetic clusters [62]: those that display >99% DNA similarity as compared to the *B. melitensis* 16M reference genome and therefore recognized as the “classical” species, and those displaying a DNA similarity close to ~97%, recognized as the “non-classical” *Brucella* species (Figure 2). While the former cluster is a homogenous group limited to mammals, the latter group is continuously increasing its members and includes more diverse brucellae strains found in various vertebrates, including mammals, Anura, fish, and chameleons [44,61,62,63,64,65,66]. The “non-classical’” *Brucella* species found in amphibians and fish exhibit high genetic flexibility are motile, fast growers, and more resistant to high acidity and unfavorable environmental conditions [61,67].

Notwithstanding the increasing availability of hundreds of *Brucella* WGS, they still seem insufficient to reveal the basis of their host restriction and differential virulence [23]. For example, among the classical brucellae, the closer phylogenetic relative of *B. canis* is *B. suis* biovar 4, with only 253 SNPs differences [68], and of *B. abortus* is *B. melitensis*, separated by ~2400 SNPs. However, these bacteria display distinct host preferences and virulence [69]. Moreover, *B. ceti* is an aggressive pathogen for cetaceans, while *B. pinnipedialis* is a mild pathogen for seals and walruses [70,71,72]. For comparison, *Salmonella enterica* serovars responsible for gastrointestinal and systemic disease [73] often have a broad host range, like *S. enterica* serovar Enteritidis [74]. However, serovars Paratyphi A and Typhi are restricted to humans, while serovar Gallinarum is a pathogen of chickens [73,74,75]. When compared, 4810 SNPs separate the genomes of serovar Paratyphi A isolates [76], and the genomes sizes of Paratyphi and Typhi differ by 14.5 kb. Nevertheless, they share the same host preference [74,75]. The genome size difference when comparing the serovar Gallinarum to Typhi is 150.3 kb, being the former smaller [74].

The similitude among the various classical *Brucella* species represents a clinical diagnostic problem since the traditional bacteriological and molecular identification methods lack resolution power [58,69]. For this reason, many *Brucella* isolates recovered from various animal species had been misclassified. One relevant example is depicted by the discovery of *B. neotomae,* a parasite of wood rats, as a zoonotic agent of two cases of human brucellosis [78,79]. Initially, this bacterium was misclassified as *B. abortus* by classical bacteriological methods. The correct identification of this species was only possible by detailed phenotypic and genotypic analysis, including WGS [78]. After these reports, investigators confirmed the infective potential of *B. neotomae* in mice [80,81]. Another example of misidentification was the *Brucella* sp. BCCN84.3 isolated from the testes of a Saint Bernard dog with orchiepididymitis. This strain displayed distinctive biochemical properties and a unique SDS-PAGE protein profile [82]. This isolate was initially identified under experts’ hands as *B. melitensis* biovar 2, afterward as an atypical *B. suis* [77]. MLVA-16, WGS, and detailed phenotypic characterization identified the *Brucella* sp. BCCN84.3 as distinct species, after 35 years of its isolation [77]. Due to their broader metabolic alternatives, non-classical *Brucella* isolates from humans were initially identified as *Ochrobactrum* spp. [83]. Likewise, *Brucella* isolates from a dwarf sperm whale (*Kogia sima*) were not successfully identified by biochemical profiles or traditional molecular typing techniques, such as Bruce-ladder PCR [84]. Still, both MLVA-16 and WGS based phylogenetic reconstruction clustered this strain with isolates of the sequence type (ST) 27, a genotype that had been previously associated with infection in humans [85]. Similar to these cases, *Brucella* isolates from rodents in Queensland, Australia, that displayed particular biochemical properties were considered atypical *B. suis* biovar 3 and kept frozen since 1965 [86,87]; only after the genomic era, the phylogenetic analysis revealed their close relationship with the non-classical BO1/BO2 clade, representing a distinct lineage [88].

## 5. *Brucella* Speciation Bottleneck Through the Domestication of the Preferred Host

Domestication has promoted inheritable phenotypic and behavioral changes in originally wild animals. These changes are linked to modifications in the genetic flow of domesticated populations, mainly as a consequence of selection for anthropogenic purposes such as (i) maximization of the number of productive offspring obtained by breeding; (ii) maximization of the efficiency of food conversion of animal products and working force, (iii) minimization of energy expenditure and losses induced by infectious, metabolic diseases, and stress [89,90,91]. The geographic and social environment, the size of the herds, the type of reproduction strategy, the proximity to wildlife, the quality of food, transport, human taste, and possibly vaccines that do not prevent transmission are selective factors for specific genes [91,92,93].

In asexual bacterial populations with limited genetic variation, epigenetic plasticity may be a significant source of variation, enabling fast adaptation to new environments, which is then fixed through regular genetic modifications. However, domestication is not unidirectional, and both the microbiome and pathobiome are selected during this process that works as a “bottleneck,” reducing the genetic diversity commonly found in wildlife populations [94]. In the case of *Brucella* organisms, we can assume that it is not a random event that the most virulent *Brucella* species, with the broader host range, are those that preferentially infect domestic animals [12] (Figure 3). This event is not a rare phenomenon. After domestication, bacterial pathogens may be selected and jump back and forth between bovids and humans during the early Holocene [95]. This hypothesis is commensurate with the less zoonotic potential of *Brucella* species isolated from wild animals, such as dolphins, walrus, or voles [12]. These observations may still be biased since comparisons between *Brucella* isolates from wildlife and domestic animals are lacking. One important exception is *B. ovis*. This bacterium is non-zoonotic and displays a strong host preference for rams, a phenomenon that demonstrates adaptation towards its host accompanied by a significant genome degradation as compared to other *Brucella* species [12,52,96,97].

## 6. *Brucella* Host Distribution from Different Geographic Areas

Despite the high genetic similarity among the various *Brucella* species, DNA analyses have shown clustering patterns consistent with the geographic origin of the preferred host [20,21,27,98]. This clustering is appreciated in brucellae isolates from marine mammals (e.g., *B. ceti* and *B. pinnipedialis*), as shown by MLST, MLVA, and WGS. The identified genotypes follow strong correspondence with their host geographic origin [27,98,99,100,101]. WGS SNPs phylogenetic reconstruction supported and extended the topology resolved by MLVA-16. The MLVA-16 “A2” and “B” clusters, however, appeared as part of a single genomic lineage, a picture that may change when more genomes are available for analysis. Likewise, the MLST schemes divided *B. ceti* into two main sequence types (ST): the ST23 complex and ST26 complexes. These two STs have porpoises and dolphins as preferred hosts, respectively. *B. pinnipedialis* ST53 and ST54 infect preferentially hooded seals, while ST24, 25, 51 and 52, infect other pinnipeds [21,99,102]. 

Additional to the genotypic clustering pattern seen in *B. ceti* and *B. pinnipedialis*, specific phylogeographic signatures have been described [27], meaning “genomic elements that show a segregation pattern that is consistent with geographic distributions of individuals” [103], constituted by differential pseudogenes, number and position of IS*711*, and by reordering and inversion of GIs and SARs.

In *B. suis*, there is a tight congruency between genotype and biovar designation [21], which differs according to geographic distribution and host range [22]. *B. suis* biovars 1 and 3 are zoonotic, and the preferred hosts are domestic pigs and occasionally wild boar [104,105]. The *B. suis* biovar 1 is common in South America and Asia [106,107,108], while biovar 1 and biovar 3 are mainly present in the United States, Australia, and China [109,110,111,112]. *B. suis* biovar 2, mainly restricted to hares and wild boars in Euroasia, also infects domestic pigs [22,113], but rarely humans [114]. *B. suis* biovar 4 is a pathogen of reindeer and caribou and a zoonotic risk in the Arctic region [105,115]. *B. suis* biovar 5 from Eastern European rodents seems confined to their natural hosts and is seldom found in domestic animals or humans [69,105,116]. 

Similar to the biovar distribution, two genetically divergent major *B. suis* clades were described by MLVA, MLST, and WGS phylogenetic reconstruction: one included *B. suis* biovar 2 and the second one *B. suis* biovars 1, 3, and 4 and *B. canis*. *B. suis* biovar 5 forms a separated branch [21,98].

According to MLST, the *B. abortus* isolates globally comprise three major clades: A, B, and C. The A and B clades include isolates from Africa and consist predominantly of biovar 1, 3, and 6. In contrast to the geographical restriction of clades A and B, isolates of clade C have a global distribution [21]. The WGS analysis supported this continental segregation [117]. However, comparisons of different bacteria from smaller geographic areas by WGS and MLVA are not always straightforward, as it will be further discussed [26,117,118,119,120].

## 7. *Brucella* Speciation and Host Preference: Small Genetic Differences Matter

When comparing *Brucella* genomes, it is essential to consider that most of their variability is found in ~3% of their content. Strains or isolates from the same outbreak or within the same host show less variation among them. This characteristic may be trivial compared to other Gram-negative bacteria such as *E. coli* and *S. enterica*. Indeed, isolates of *S. enterica* serovar Typhimurium can vary from 2% to 20% only in genome size; also *E. coli* O157:H7 has a 20% larger genome than *E. coli* K-12 [11,121]. The variation found within those genera far exceeds the *Brucella* genomic diversity [11]. However, it is remarkable that despite this homogeneity, the *Brucella* species display different phenotypic characteristics, host preference, virulence, and zoonotic risk. Some elements that might account for genome variability are related to (i) VNTR, (ii) differential SNPs positional patterns, (iii) an increased number of pseudogenes as compared to the most common recent ancestor, (iv) number and distribution of mobile genetic elements (e.g., insertion sequences), and (v) number and distribution of GI and SARs [11,20,27]. Although there is no investigation of the epigenetic modification sites in *Brucella* organisms, the CcrM methylase that recognizes the double-stranded sequence GANTC may induce the methylation on both strands and therefore be a source of epigenetic modifications [122,123].

### 7.1. Variable Number of Tandem Repeats

DNA tandem repeats (TRs), also called satellite DNA, are inter- or intragenic nucleotide sequences repeated two or more times in a head-to-tail manner. Loci that include TRs are hypermutable because tandem tracts are prone to strand-slippage replication and recombination events that cause the TRs copy number to increase or decrease [124,125]. The evolutionary clock speed at each locus is variable, and differences in the TR copy number at multiple loci have proven useful for gathering epidemiological information as well as characterization of intra- and inter-population variation [126,127]. Furthermore, rearrangements of intergenic TRs can confer transcriptional evolvability and phenotypic variation [124,125].

The first application of VNTRs for *Brucella* genotyping used microsatellite fingerprinting based on the number of TRs of the sequence “AGGGCAGT” at eight loci in the genome. The technique was called “HOOF-Prints” as an acronym for Hypervariable Octameric Oligonucleotide Finger-Prints [128]. Afterward, Le Flèche et al. [20] and Whatmore et al. [19] applied this method into a phylogenetic context. Le Flèche et al. [20] expanded the TRs to 15 different loci, named MLVA-15, divided into two panels: panel 1, composed of eight “user-friendly” minisatellite markers with good species identification capability, and panel 2, a complementary group of seven markers with higher discriminatory power [20]. Subsequently, the original eight HOOF-Prints were used altogether with an additional eight short 5- 8 bp TRs, to reduce the risk of a match due to homoplasy and included in an MLVA-16 scheme [20]. Whatmore et al. [19] implemented a larger scheme based on 21 different loci aiming to increase the resolution of the technique.

The use of MLVA has proved useful for tracking *Brucella* outbreaks or infection sources to study the molecular epidemiology of *Brucella* in specific countries [129,130,131,132] and in a worldwide context [98]. Despite its usefulness, the homoplasy in VNTR markers in some *Brucella* species, as *B. abortus* and *B. ovis*, lowers the levels of discrimination and resolution of cladograms as compared to WGS phylogenetic analysis [26,117,120]. To reduce homoplasy and to understand phylogenetic relationships deeper, a systematic approach in *Brucella* MLVA, is to use the most stable VNTRs or provide further weight to the markers [133]. However, this could limit the evidence of more recent epidemiological connections [134]. Species like *B. melitensis* and *B. ceti* have proven a fair comparison of MLVA-16 cladograms and WGS phylogenetic reconstruction in the global context [27,135]. Nevertheless, MLVA-16 lacks resolution in comparative studies between *B. melitensis* and *B. abortus,* and inferring phylogenetic distances, clusters or groups within the same location is not as straightforward as in WGS-based approaches [26,136].

### 7.2. SNPs Barcode Patterns and Allelic Variation

SNPs have simple variation patterns, low mutation rates, and low levels of homoplasy [137]. Halling et al. [138] used SNPs analysis for the first time in *Brucella* to compare the three available genomes at the moment: *B. abortus* st. 9-941, *B. melitensis* st. 16M and *B. suis* st. 1330. They found an average of one SNP per 463 nucleotides [65,138]. SNPs positions across *Brucella* genomes relative to phylogenetic reconstruction can be interpreted as distinctive barcode patterns, including not only information regarding the SNP itself but also its positional context in the genome. Some specific SNPs clusters resembling barcodes or fingerprints were identified in *Brucella* genotypes from terrestrial and marine mammals [27]. This differential pattern may be relevant since SNP variations could influence the expression of neighbor coding sequences (CDS) or RNA coding genes, contributing to niche adaptation [103,139].

On the other hand, one SNP could be the cause of host adaptation and increased virulence as described for other bacteria, despite being located in intergenic regions [140,141]. Isolates of *B. abortus* collected during a time-span of three years and from a very restricted geographical area of 18.75 Km^2^ showed 1–18 SNPs that allowed phylogenetic sub-clustering [117]. *B. melitensis* isolates from a restricted geographical area in South-East France from three different hosts displayed a single SNP difference [142].

MLST schemes are valuable tools to investigate genetic relationships within species of the *Brucella* genus [8,21]. The MLST method uses alleles as the unit of comparison, rather than nucleotide sequences; in this sense, each allelic change is counted as a single event, regardless of the number of nucleotide polymorphisms involved. Each locus is part of a profile or an ST designated by a number [143]. The corresponding ST may be associated with biological properties, such as virulence or host preference [144]. The first MLST profile for *Brucella* was developed in 2007 and included nine distinct genomic fragments, seven representing classic housekeeping genes known as *gap*, *aroA*, *glk*, *dnaK*, *gyrB*, *trpE*, and *cobQ*. The two remaining loci were variable markers: a fragment of the *omp25*, encoding a 25 kDa outer membrane protein, and *int-hyp* region, which included an intergenic segment and, partially, a hypothetical protein [8]. As the technique developed and the *Brucella* genus expanded, further variation was obtained, and the original protocol improved the resolution and discriminatory power. After that, the 9-loci scheme incorporated 21 loci, including multiple alleles for loci [21].

The current MLST profiles have shown diverse genotypes among samples of the same species, unveiling variability related to geographic origins, phenotypic biovars profile [21], pathogenic potential, and host restriction [99]. As stated before, although in some cases the MLST classification seems a practical taxonomical tool, the genomic variability should not be restricted to types, but rather to phylogenetic relations; therefore, conclusions regarding the dispersion and evolution of *Brucella* should be based on ancestor/descendant relationships and not on MLST.

### 7.3. Pseudogenes

The importance of pseudogenes in bacterial diversification and evolution is just recently being unveiled. There seems to be a consensus within the scientific community on the functional definition of what is a pseudogene. At the same time, there are different criteria related to which are mechanisms of pseudogenization and how they are inferred from automated annotation pipelines, or manually annotated [145]. This is not trivial, since conclusions based on pseudogene analysis might differ from one study to another, simply because the criteria for pseudogene annotation are different. From the functional point of view, pseudogenes are “fossil” sequences, which may serve as “genetic reservoirs” for plasticity, required for evolution [145,146]. In other cases, they are genes with a new function, since they participate in gene regulation and RNA interference (RNAi) roles [147,148]. From the mechanistic point of view, bacteria pseudogenes are defined here as any gene containing deletions, insertions, or both, which remove start or stop codons, or at least one in-frame stop codon that may or may not involve frameshifts compared with orthologs, possessing coding or regulatory fragment sequences. Some of the most common mechanisms found after manual curation of eight *Brucella* genomes are depicted in Figure 4 and Appendix A. Analysis of *Brucella* genome degradation has been studied previously, using different pseudogene definitions and methods for data extraction [42,53,138]. *Brucella* species show a variable number of pseudogenes, and most of them are in the smaller chromosome [27,42,96]. This event is not unexpected, since the smaller *Brucella* chromosome is generically known as “chromid” probably derived from a megaplasmid possessing a significant number of dispensable non-essential genes than the larger chromosome, which retains most of the housekeeping genes [3].

Although *B. ovis*, the species with the largest number of pseudogenes, is not a zoonotic species, it is a relevant pathogen for rams and highly specific for its host [42]. After *B. ovis*, *B. ceti* is the second species with a higher number of pseudogenes (Appendix A). The number of pseudogenes in *B. ceti* and *B. ovis* also supports the idea that members of the genus *Brucella* have undergone genome reduction according to the “domino effect,” in which pseudogenization of one gene may induce the failure of linked functions and the concomitant loss or deletion of more genes by genetic drift [67,146,149]. As previously stated, *B. ceti* also shows a strong host preference.

Phylogenetic analysis from whole-genome sequences of *Brucella* isolated from marine mammals shows that they represent a branch separated from isolates obtained from terrestrial animals (Figure 2 and [27]). Analysis of the extent of gene degradation at nodes of a phylogenetic tree allowed the finding of putative primary events targeting specific metabolic pathways that have become fixed in this population. The isolates from marine mammals diverging from those isolates of terrestrial animals have lost functions related to energy metabolism, amino acid transport, metabolism, and gene regulation. Extensive pseudogenization in pathways related to fatty acid metabolism, in particular, loss of function of an acetyl-CoA acyltransferase and an acetyl-CoA C acetyltransferase, very likely impairs fatty acids synthesis and beta-oxidation in these isolates. There was further genome degradation in the *B. ceti* strains from the Pacific, Mediterranean, and Atlantic dolphins. In marine brucellae, amino acid catabolism and pyruvate fermentation seem irrelevant for survival. The genome decay pattern suggests that the loss of function is not a stochastic event but may follow a bottleneck selection by the cell host environment, favoring adaptation and, in course, the generation of ecospecies in marine mammals [27,99]. These properties also seem present in some of the Anura *Brucella* strains [150].

In course, this loss of metabolic pathways and functions no longer required may promote the facultative or obligate parasitic bacterial stage, suggesting that pseudogenization may narrow the host range and vice versa.

### 7.4. Genetic Mobile Elements

The IS transposition can induce gene inactivation, constitutive expression, or repression of adjacent genes, by reordering IS promoters or terminator sequences. They can also promote the inversion, deletion, duplication, and fusion of replicons if multiple copies of an identical ISs spread over the genome [151,152]. Three main ISs or GMEs described in *Brucella* are the IS*711*, Tn*2020,* and Tn*1953* [153,154]. The IS*711* described initially in *B. ovis* has an 842 bp length, a G + C content similar to the rest of the *Brucella* genome and is delimited by 20 bp imperfect inverted repeats [153]. The Tn*2020*, of 6 002 bp length, was discovered in *B. abortus* and codes three polypeptides with apparent molecular masses of 71 kDa, 22 kDa, and 14 kDa [154]. The Tn*1953*, an 18kbp region discovered in the second chromosome of *B. suis* 1330, shows similarity with the anomalous region IncP. The Tn*1953* is also present in *B. canis*, *B. neotomae*, *B. ceti,* and *B. pinnipedialis*, but absent in *B. melitensis*, *B. abortus,* and *B. ovis* [155]. Among these IS, only IS*711* is considered a specific element of the *Brucella* genus and appears to be the only active transposable element in *Brucella* [155,156]. The number and positions of IS*711* in the *Brucella* genomes (Table 1) serve as a fingerprint for diagnosis and species identification [157,158,159,160]. The transposition of IS*711* has occurred several times in *B. abortus*, *B. ovis,* and *B. pinnipedialis* genomes [156,160]. This genetic element plays a role in genome degradation and host adaptation, and it is commensurate with the host geographic distribution [27,42].

In *B. abortus*, the IS*711* induces variation at the strain level as demonstrated by three major deletions in the genomes of three different bacterial stocks of the same *B. abortus* strain kept in different laboratories: (i) 2308 (NC_007618 and NC_007624), (ii) 2308A (GCA_000182625.1), and (iii) 2308 Wisconsin (ERS568782). The deleted sequences are enclosed by repetitive elements and by IS*711* [163]. This event could have induced the loss of regions in 2308 and 2308A as compared to 2308W. Therefore, the conventional concept of strain definition needs to be revised, since variation can be found even among *Brucella* reference strains. Recently, an intact prophage induced by mitomycin has been identified in *B. innopinata,* named BiPBO1, of 46,877 bp length and coding eighty-seven putative gene products. Although this prophage was present in *B. melitensis* and *B. abortus*, its expression could not be induced by the same method, followed by *B. innopinata* [51]. This characteristic suggests a stage of pseudolysogeny [164] since it is likely that several *Brucella* organisms do not encode all proteins required for the assembly of intact particles [51]. Since this prophage is also present in *Ochrobactrum*, it seems ancestral in the several Alphaproteobacteria [51]. The prophage could be expressed in “non-classical” brucellae but not in the classical strains, a fact that corresponds with the closer phylogenetic relation of the former group with *Ochrobactrum* [51].

### 7.5. Genomic Islands and Anomalous Regions Modifications and Distribution

The GIs is a term that usually refers to DNA sequences acquired during evolution, and so, they display some differences (e.g., G + C content) with the core backbone genome composition. The GIs may be considered as MGEs because they show genetic instability and nearby frequently have repetitive elements, mobility genes (transposases, integrases), and tRNA genes [165,166,167]. SARs share anomalous regions that contain syntenic protein-coding genes in different *Brucella* genomes as given by the OrthoMCL ortholog data and double-checked by BLAST2seq [96]. There are 24 GIs and SARs described in the *Brucella* genus (Appendix A, [27]). Some of them are shared among various species, while others are species-specific. *Ochrobactrum* genes encoded in the GIs suggest that the horizontal acquisition occurred at the *Brucella/Ochrobactrum* ancestor and lost in the emergence of *Ochrobactrum* [43], but kept in *Brucella*. *Brucella* GIs IncP, GI-1, GI-2, GI-3, SAR1-2, and SAR1-5 include a full-length *int* gene. Some of the *int* genes are similar to those that belong to the tyrosine recombinase superfamily (encoded in SAR1-5). Other genes show characteristics similar to those of P4-phage integrases that mediate the excision and transfer or both, of GIs (encoded in IncP and GI-3). These genes, and likely other accessory proteins, mediate GIs excision and rearrangement [60].

There are variations in the position and orientation of genes included in the GIs as revealed by *B. ceti* and *B. abortus* genome comparisons [27,117]. The GIs and SARs were consistent with each one of the species, without significant deletions or insertions. However, there is a particular reordering pattern within the isolates of the same species [27,117]. This result is relevant since different positions of the chromosome regions can be associated with different expression patterns [167]. Environmental stimuli can affect the expression level of *int*, and then mediate the GIs excision and reordering phenomenon. The *Brucella* GIs instability suggests that these genomic elements are still undergoing an integrative process and raise questions about their role in host specificity, phenotypic differentiation, and virulence [27,60].

## 8. Microevolution of *Brucella* in the Hosts

Bacterial microevolution may rise in “clonal” infections, like tuberculosis or brucellosis. This event may occur when variants are detected within a single host or in related hosts during a single outbreak [168]. Studies in *Brucella* have reported variation of SNPs among isolates recovered from the same host tissues or among isolates obtained from the same host sampled at different times. A study of a *B. canis* outbreak in Hungary reported variation in five of 15 different VNTR alleles at multiple tissues and dates, during three months, suggesting rapid genetic changes in *B. canis* that produced the emergence of alternative alleles at the VNTR loci [169]. Another example using MLVA is provided by Maquart et al. (2009) in marine *Brucella*: they cultured and typified bacteria from more than one tissue in 69 marine mammals, and they observed more than one genotype in sixteen of them. Moreover, *B. suis* 01-5744 recovered from four pigs, after 63 days of experimental infection, showed VNTR variation in three of 21 loci, as compared to the original strain [19].

Genomic comparison of isolates recovered during a specific period has also reported SNPs and length variation in *B. suis* and *B. melitensis*. For example, genomes of two *B. suis* clinical isolates from the same patient, eight years apart, showed a length difference of 2757 bp [170]. Ke et al. (2012) performed an in vivo experiment where they looked for genomic changes in *B. melitensis* 16M recovered from infected mice at different times, simulating acute and chronic infections. This study reported 11 SNPs during the first week, and 5 019 SNPs at the 13th week [171]. Even intergenic single SNPs can dramatically impact the phenotype of a bacterium [140,141]. The variation within the host reinforces the need for further research to confirm each SNP role, or the role of the some SNPs combination, on the *Brucella* phenotype, including attenuation. Indeed, looking at these variations requires high-quality metadata and high-resolution power techniques with high accuracy. 

## 9. Concluding Remarks

Most of the emergent human pathogens have a zoonotic origin where transgression of host barriers is critical. Colonization of new environments requires behavioral adjustments in a bacterial population, which could lead to stable genetic modifications. In terms of understanding the emergence of pathogens, how bacteria living in soil can eventually adapt to milieus as different as an intracellular environment is an intriguing and relevant question. 

Intracellular clonal bacteria that probably arose when their ancestor adapted to a particular host are confronted with the challenge of keeping genetic variability to respond to continuous stimuli originated by the host, especially, the immune system. With little chance of recombination or HGT, a bacterium with a medium-size genome, such as members of the *Brucella* genus, has limited sources for genetic plasticity.

The analysis of *Brucella* genomes from domestic and wildlife animals has expanded the knowledge of *Brucella* genetic variability. Lineages of divergence according to the geographic origin in *B. ceti* and *B. abortus* become evident through the (i) specific IS*711* insertion patterns across the genome, (ii) differential IS*711* number, (iii) specific SNPs signatures across phylogenetic clusters, and (iv) pseudogenization of metabolic pathways. In contrast to the genomes of *Brucella* from marine mammals, the genomes of domestic animal bacteria have fewer SNPs and pseudogenes, suggesting a reduction of diversity in the latter bacteria as a consequence of domestication. Likewise, the diversity and divergence among the non-classical brucellae seem more extensive than the classical counterparts. This event is not an unexpected outcome, as the domestication process itself works as a selective pressure in the gene flow as a consequence of the co-evolution of pathogens and hosts [12,172]; this could have promoted and selected the close similarity observed in the *B. abortus* genomes isolated from cattle. Similar events likely occurred in the case of *B. melitensis* and *B. suis* with their respective hosts. We require more genomes of classical and non-classical organisms from wildlife animals for establishing comparisons. According to host specificities, other signatures of adaptation, including the fine-tune regulation of gene expression, are likely to occur in this intricate liaison. High-resolution studies of both bacterial population genetics, as well as studies of specific bacterial clusters displaying particular phenotypic traits, are necessary to unveil microevolution events that might be occurring. Despite the current knowledge, the central questions of brucellosis, regarding host adaptation and virulence variation require a solution. The fact that *Brucella* from wood rats, hares, and cold-blooded animals can infect humans [78,114,173] depicts the zoonotic potential of all members of the genus and the ability of these bacteria to adapt from the preferred host to other animals and cause disease.

## Figures and Tables

**Figure 1 ijms-21-07749-f001:**
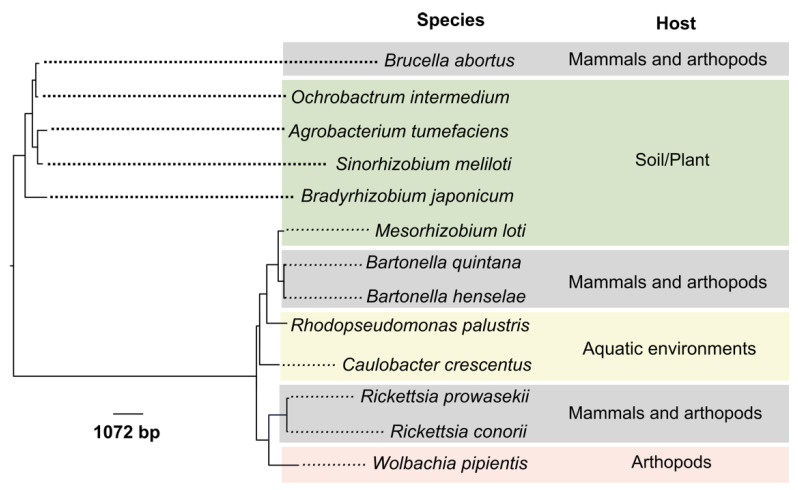
The phylogeny of some representative Alphaproteobacteria inferred by maximum likelihood reconstruction based on the 16S rRNA gene. The sequences used as outgroup [14] were trimmed out from the tree to enhance the resolution. The analysis involved 17 nucleotide sequences; there were a total of 1191 positions in the final dataset. Appendix A indicates the details of the strains and sequences used for the analysis. *Ochrobactrum intermedium* may behave as opportunistic bacteria in immunocompromised human hosts [15,16].

**Figure 2 ijms-21-07749-f002:**
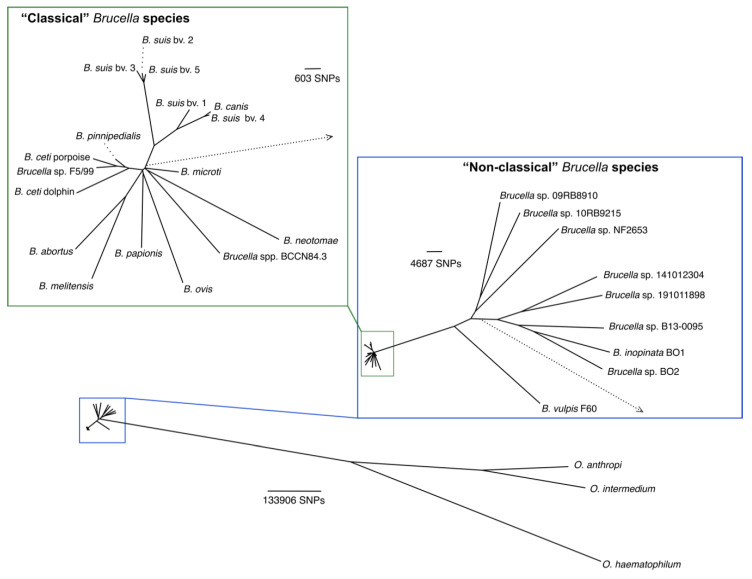
Phylogenetic relationship of “classical” and “non-classical” *Brucella* species and *Ochrobactrum* sp. based on 669529 SNPs. Segments of the tree were magnified to increase resolution. The SNPs scale is next to each magnified region. The dotted arrow represents the branch linking the magnified regions to the whole tree. A blue square highlights all *Brucella* species; the “classical” species are within the up left square. Modified from [77]. In Appendix A are the details of the strains and sequences used for the analysis.

**Figure 3 ijms-21-07749-f003:**
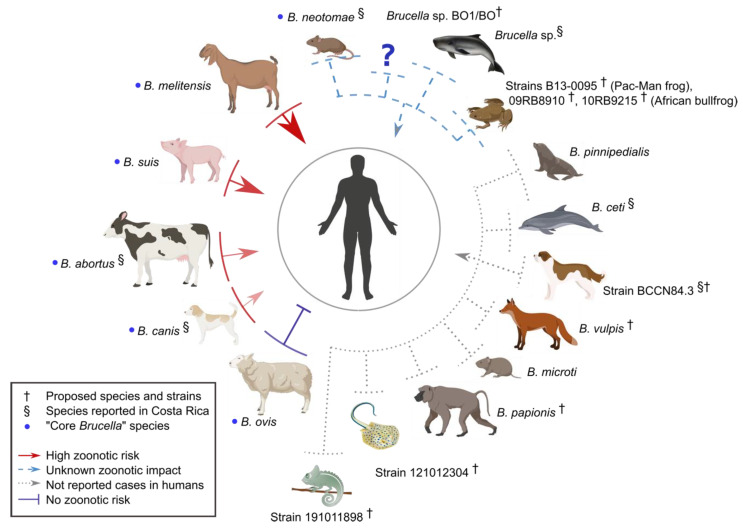
Zoonotic potential of the various *Brucella* species. The colors and arrow sizes represent the zoonotic risk displayed by each species isolated from its preferred host. Appendix A presents a detailed analysis. Created with Biorender.com.

**Figure 4 ijms-21-07749-f004:**
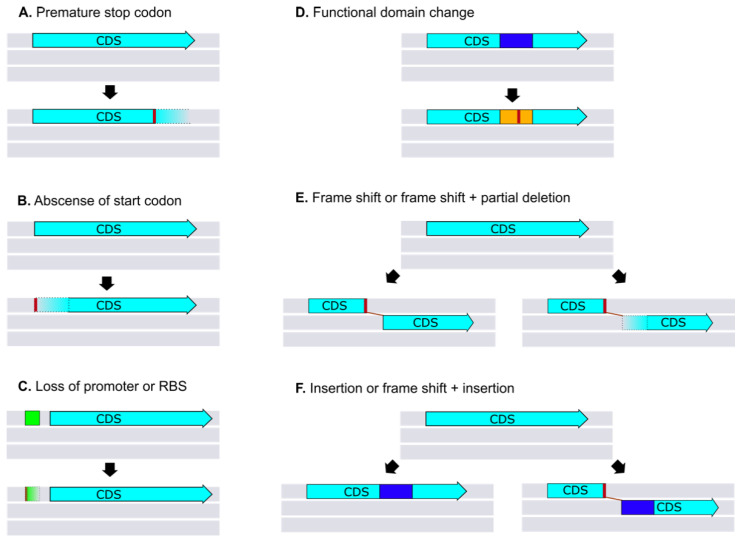
Pseudogenization mechanisms. Schematic representation of pseudogenes induction found in *Brucella* after manual curation of genome data (Appendix A). (**A**–**C**) show pseudogenization induced by polymorphisms that change the length of the CDs by the acquisition of a premature stop codon (**A**), loss of the start codon (**B**), or alteration of gene expression by promoter loss or regulatory binding sites (**C**). Polymorphisms also can induce a functional domain change (**D**) that results in a different product or frameshift (**E**). Deletions and insertions, shown in (**E**,**F**), can also induce frameshifts.

**Table 1 ijms-21-07749-t001:** IS711 number according to “classical *Brucella*” species.

Species	Copy Number	Reference
*B. abortus*	6 complete, 1 truncated	[138]
*B. ceti*	>20 complete	[27,161,162]
*B. melitensis*	7 complete	[156]
*B. ovis*	38 complete	[42]
*B. pinnipedialis*	>20 complete	[161,162]
*B. suis*	7 complete	[156]

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
