# Peer review of "Brucella Genomics: Macro and Micro Evolution"

_ijms, 2020, doi:10.3390/ijms21207749_

Round 1

Reviewer 1 Report

The manuscript "Brucella Genomics: Macro and Micro Evolution" is a review reporting the "state of the art" about the current knowledge about Brucella species diversity looking at the evolution driven by host adaptation and environment and how much small genetic differences matter.

The idea is good and provides some important pieces of information for specialists but comes across also to scholars in different fields interested in relationship between living organism and the whole environment.

However, this reviewer recommends some major revisions.

1) At the paragraph 4 (starting from the line 164) the authors write about taxonomy trying to explain what kind of "exercise" is needed being taxonomy depending on two dialectically components. It is obscure to this reviewer the whole meaning of this very long discussion considering it beyond the focus of the review. All the concepts and the citations that the authors write from the line 165 to the line 196 seem to be the focus of another review about how to classify bacteria and what the problems associated to taxonomy are. This reviewer recommends to delete the entire part (line 165-196) and let the paragraph start from line 197.

2) line 202: the sentence starting from "The bacteria found in amphibians..." does not sound; there could be a mistake: could it be "shown" instead of show ? Please, check and revise the sentence.

3) line 212-218: again the paragraph is beyond the focus of the review. The comparison made by authors takes the reader out of the main focus and does not add any important information

4) Paragraph 7.2 ("SNPs patterns and allelic variation")

Does the title has to be SNP pattern and allelic variation ?

Then and more important, the paragraph describe how SNIPs can be the cause of host adaptation and how many SNPs in B. abortus have been found in some geographical ares  (costa Rica, south east France etc.). There is no mention about Italy where Brucellosis is endemic and constitutes a seriuos threat for health of humans and animals and also affect the economy of the South of Italy which produces peculiar cheese using buffalo milk. 

Two papers report SNPs which are related to B. abortus biovar: Borriello G. et al Appl. Environ. Microbiol. 2013, 79, 1039–1043; Borriello et al. Biomolecules 2020, 10, 788.

Of course the authors don't have to cite the article indicated if they have better to cite to describe the situation in Italy, but it is important to describe Brucella biovars and their host preference in South of Italy.

Minor points:

Hoping to be helpful for the authors, I'm going to indicate some typos and/or sentences which are not clear:

1) line 36: Does "which" has to be substituted with  "whose" ?

2) line 48: "The Brucella genome is, conserved..." coma after "is" has to be eliminated

3) line 290: "Although the MLST classification, may have some pratical..." coma after "classification" has to be removed.

4) line 292: "as same as"; is it better to say "as well as" or "just as" ?

5) line 374: I would add "are" at the end of the sentence

6) line 380: "...Brucella MLVA is to use only the most stable..."; Could it be better "Brucella MLVA is to use only in the case of the most stable..." ?

I finally suggest to specify the meaning of acronyms:

SNP; IS; SAR. Just to be more readable for non-specialists

Author Response

 "Please see the attachment, table with the response to reviewer #1, thanks!

Reviewer 2 Report

In general, this is an interesting and complete review which covers the subject from a new angle.  This reviewer does, however,  believe that there are several conceptual errors that need to be addressed.

The manuscript is, also over long and rambling, and could be made more concise and to the point by cutting/reducing certain sections and reorganizing the order in which the ideas are presented. The reference section is excessively long (185 references) for such a short review. This should be shortened. Only published articles (not those in revision) should be cited.

Examples of sections to shorten are the first two paragraphs, and the figure on pseudogenes (copied from another paper).

For idea order; The manuscript introduces concepts such as MLVA and MLST and SNPs before describing them.   It would be better to start with a brief description of the different methods and present them in increasing sensitivity…so MLST first, then VNTR followed by WGS.  Although it is now old fashioned and rarely used, it would be good to mention PFGE as this clearly shows how each species and biovar has a unique genome organization.

L463  After B. ovis, B. ceti is the second species with a higher number of pseudogenes.  You don’t introduce the idea that B ovis has the highest number of pseudogenes.

Specific Points

L141  The VirB system was NOT retained and remodeled. They were acquired at different times from different sources.

Paragraph l257 onwards.

Suggests that virulence for humans has been selected.  This hypothesis ignores the cases of B suis bv4 and bv3 which are highly virulent in man, but rarely encountered in human infections.  This is because, unlike B. melitensis and B. abortus, they are not so common in farm animals, meaning that there is less chance for humans to get infected.

L 573-5 ‘Likewise, the diversity and divergence among the non-classical brucellae seem more extensive than the non-classical counterparts. This event is not an unexpected outcome, as the domestication process itself works as a selective pressure in the gene flow [102] that could have induced the close similarity observed in Brucella genomes isolated from domestic hosts.’  This reference does not address bacterial or viral pathogens at all, and is mainly dedicated to resistance to plant pests such as insects. The authors appear to suggest that prevalence of a specific genotype suggests selection, however, if a specific genotype is associated with a specific host, the prevalence of a genotype will reflect the prevalence of the host. Therefore, as B. melitensis and B. abortus are associated with domestic animals and have been spread across the world with livestock.  It is not a reduction in diversity, but an increase in incidence of a highly stable genome.

L296-8.  the closer phylogenetic relative of B. canis is B. suis biovar (bv.) 3.   This is not what Fig 2 shows. Please clarify.

L447  The discussion of the paper by Ke et al should be removed.  The second strain in this paper is NOT B. melitensis.

Author Response

Please see the attachment with the table containing the response to reviewers, thanks!

Round 2

Reviewer 1 Report

The author made the essential changes required. The manuscript can be accepted in the present form